# The topology of genome-scale metabolic reconstructions unravels independent modules and high network flexibility

**Verónica S. Martínez**[1,2☯], **Pedro A. Saa**[3,4☯], **Jason Jooste**[1], **Kanupriya Tiwari**[1], **Lake-Ee Quek**[1,5], **Lars K. Nielsen**[1,2,6,7]*

**1** Australian Institute for Bioengineering and Nanotechnology (AIBN), The University of Queensland, Brisbane, Queensland, Australia, **2** ARC Training Centre for Biopharmaceutical Innovation (CBI), Australian Institute for Bioengineering and Nanotechnology (AIBN), The University of Queensland, Brisbane, Queensland, Australia, **3** Departamento de Ingeniería Química y Bioprocesos, Escuela de Ingeniería, Pontificia Universidad Católica de Chile, Santiago, Chile, **4** Instituto de Ingeniería Matemática y Computacional, Pontificia Universidad Católica de Chile, Santiago, Chile, **5** The Charles Perkins Centre, School of Mathematics and Statistics, The University of Sydney, Sydney, Australia, **6** Metabolomics Australia, The University of Queensland, Brisbane, Queensland, Australia, **7** The Novo Nordisk Foundation Center for Biosustainability, Technical University of Denmark, Kgs. Lyngby, Denmark

☯ These authors contributed equally to this work.
* lars.nielsen@uq.edu.au

**Data Availability Statement:** All relevant data are within the manuscript and its Supporting Information files.

## Abstract

The topology of metabolic networks is recognisably modular with modules weakly connected apart from sharing a pool of currency metabolites. Here, we defined modules as sets of reversible reactions isolated from the rest of metabolism by irreversible reactions except for the exchange of currency metabolites. Our approach identifies topologically independent modules under specific conditions associated with different metabolic functions. As case studies, the *E.coli* *i*JO1366 and Human Recon 2.2 genome-scale metabolic models were split in 103 and 321 modules respectively, displaying significant correlation patterns in expression data. Finally, we addressed a fundamental question about the metabolic flexibility conferred by reversible reactions: "Of all Directed Topologies (DTs) defined by fixing directions to all reversible reactions, how many are capable of carrying flux through all reactions?". Enumeration of the DTs for *i*JO1366 model was performed using an efficient depth-first search algorithm, rejecting infeasible DTs based on mass-imbalanced and loopy flux patterns. We found the direction of 79% of reversible reactions must be defined before all directions in the network can be fixed, granting a high degree of flexibility.

## Author summary

Genome-scale metabolic reconstructions represent all biochemical reactions that an organism can accomplish. These reconstructions are complex and often difficult to study in great detail. A way to overcome this limitation is to focus on specific pathways or subsystems. We present a novel method to identify metabolic modules based on the network topology. The method relies on reaction directions and ignores currency metabolites,

**Funding:** VSM is supported by the Advance Queensland Women's Research Assistance Program (WRAP213-2019RD1). PAS acknowledges the support from ANID through Fondecyt de Iniciacion (Grant No 1190074) FONDAP through the Center for Genome Regulation (CGR) (Grant No 15090007), and the National Center for Artificial Intelligence CENIA FB210017, Basal ANID. KT is the recipient of a University of Queensland International Fees and Living Scholarship. LKN is supported by the Novo Nordisk Foundation (Grant No NNF14OC0009473 and NNF20CC0035580) and the Australian Research Council (Grant No IC160100027). The funders had no role in study design, data collection and analysis, decision to publish, or preparation of the manuscript.

**Competing interests:** The authors have declared that no competing interests exist.

which artificially connect distant metabolic reactions. In this way, topologically independent modules are built, where inputs and outputs are controlled by irreversible reactions. The method is automatic and unbiased, and, the result is a set of condition specific modules with defined metabolic functions. As a proof-of-concept we generated biologically relevant modules for the *E.coli* and Human genome-scale metabolic reconstructions supported by transcriptomic data. Finally, we applied the novel approach to study the network flexibility conferred by reversible reactions. In the case of the *E. coli* model, we found that the direction of 79% of structurally reversible reactions (those not directionally constrained by surrounding irreversible reactions) must be fixed to determine all the reaction directions in the network. Therefore, reversible reactions operate practically independent of each other.

## Introduction

A genome-scale metabolic model (GeM) is a comprehensive mathematical representation of an organism's metabolism [1, 2]. To date, GeMs for more than 6,000 organisms, including all model organisms, have been reconstructed [3–8]. This network representation is widely employed to study the metabolic phenotype of cells with applications ranging from strain development, modelling interactions among multiple cells or organisms, understanding human diseases to the study of evolutionary processes [8–13].

GeMs describe all metabolic capabilities of an organism, i.e., all biochemical reactions that can carry flux under any condition. These detailed models contain thousands of reactions, which can confound more detailed studies of network properties and functions. A common strategy to overcome this limitation is to focus the analysis on one or a few model subsystems. Subsystems have been defined by conventional biochemical pathways in online databases such as the Kyoto Encyclopedia of Genes and Genomes (KEGG) [14] and BioCyc [15]. Subsystems have been used to map omics data [16] and for model reduction [17], yet their definition is arbitrary and identical for all organisms. Recognising the diversity and uniqueness of the metabolism in individual organisms, a more satisfying alternative would be to generate model subsystems in an unsupervised manner relying exclusively on the specific topology of the studied metabolic network.

The topology of metabolic networks has been widely studied by graph theory methods. Early work by Barabasi and colleagues concluded that metabolic networks are scale-free, hierarchical networks with highly connected modules overlapping known metabolic functions [18, 19]. However, these analyses did not consider the nature of the edges and it soon became apparent that the extremely short average pass length observed was realized through cofactors (e.g., ATP, NADH, NADPH), whereas the flow of carbon from a substrate to a product often is quite long. Following a more biologically meaningful interpretation of the network topology, by excluding currency metabolites (cofactors and moieties) and accounting for directionality of irreversible reactions, Ma [20, 21] observed that metabolic networks can be broken into a modest number of strong networks (i.e., networks where each metabolite can be reached from every other metabolite). The network arranged as a directed bow-tie structure with a substrate subset connected to a product subset through a *giant strong component* corresponding to *central carbon metabolism* [20, 22]. Another approach for inferring and studying metabolic modules/pathways is based on structural (stoichiometric) analysis [23–26]. For this task, classical Elementary Flux Modes (EFMs) has been adapted for enumerating flux patterns in metabolic subnetworks (i.e., modules) under biomass-optimal growth [23, 25], incorporating even loopless criteria [27] avoiding thermodynamically infeasible flux cycles [24]. While these approches

have yielded deep insights about the flexibility and functioning of metabolic newtorks, their applicability still remains limited to small- to medium-sized models.

This work presents a novel approach to generate topologically independent metabolic modules exploiting the network topology and directionality constraints. The *E.coli* iJO1366 [4] and Recon 2.2 [7] GeMs were subdivided in topologically independent modules and evaluated for their biological relevance under specific growth conditions. The clustering approach provides fundamental insights into the role and flexibility conferred to metabolic networks by reversible reactions. We quantitatively estimated the network flexibility by counting in each module the number of feasible Directed Topologies (DTs), which represent *consistent* flux solutions [28] where all reactions carry flux, and hence, the directions are fixed. Notably, these DTs are maximal pathways known as Flux Topes (FTs) [23], which have been recently applied for exploring the flexibility of optimal network states, and correcting thermodynamically infeasible cycles [29]. Under the assumption of 'thermodynamic' isolation, the (Cartesian) product of the DTs of the different modules provides an unprecedented upper bound estimate of the 'topological' degree of freedom of the network.

## Results

### Model reduction and compression

GeMs are large models with thousands of reactions, some of which are isolated and unable to carry flux, while others are part of linear pathways that can be compressed. As an initial step, all blocked reactions were removed and the model compressed to generate a more manageable model for clustering.

### *E. coli* iJO1366

The *E.coli* iJO1366 [4] contains 2,583 reactions (941 reversible reactions) and 2,135 metabolites (330 boundary metabolites). Under aerobic growth in medium with glucose as sole carbon source, the initial 941 reversible reactions were reduced by 74% to only 248 "structurally" reversible reactions after model reduction and compression (Table A in S1 Table). Using the network topology and original directions, flux variability analysis (FVA) [30] was performed to identify and remove blocked reactions resulting from singleton metabolites and, where possible, constrain the flux direction of active reactions. The result was the identification of 242 blocked reactions and 534 new irreversible reactions, causing a 57% reduction in reversible reactions as well as a 60% reduction in metabolites involved in those reactions. Next, the reduced model was compressed by lumping together reactions in linear pathways as they carry the same flux and are fully coupled [31]. Overall, the model reduction process led to a total compression of the model from 2,583 to 1,419 reactions (45% reduction), and from 2,135 to 971 metabolites (55% reduction) (Table A in S1 Table). More importantly, the original 941 reversible reactions were reduced to 248 (74% reduction). We denote these remaining reversible reactions as "structural" reversible reactions. Notably, there is a considerable reduction in the number of metabolites participating in reversible reactions from the initial 2,135 to just 382 (28%); if we know the concentration of these metabolites and the $\Delta_r G^{'0}$ of the structural reversible reactions, we can determine the directions of all metabolic reactions in the model [32, 33].

### Human model recon 2.2

The reduction of the human model Recon 2.2. [7] was performed following the same methodology used for the *E. coli* model. Recon 2.2 has 7,785 reactions (3,782 reversible reactions) and 6,047 metabolites (723 boundary metabolites).

After the initial model reduction by FVA, 1,878 reactions were found to be blocked and 1,007 reactions became irreversible. The total number of metabolites in the model was reduced by 36% and the reversible reactions by 52%. The compression of the model further reduced the number of reversible reactions to 1,582 structurally reversible reactions, and the number of metabolites participating in reversible reactions to 1,523, only 42% of the initial number of reversible reactions (Table B in S1 Table).

## Modular topology and clustering

Metabolism is organised into semi-autonomous modules sharing a common pool of currency metabolites (co-factors and moieties) [21], but otherwise weakly connected. Here, we considered irreversible reactions as the natural boundary between modules. Irreversible reactions are thermodynamic insulators preventing downstream products from affecting upstream reactions. Moreover, they are in many cases the "committed pathway step" under allosteric regulation, which greatly reduces the control exerted by the upstream substrate. Using irreversible reactions to define the boundaries of modules, the metabolic models were subdivided by clustering the reversible reactions by common metabolites connecting them, disregarding connections associated to currency metabolites (co-factors and moieties). As a result, we identified functional modules in the *E. coli* and human metabolism.

### *E. coli* iJO1366

The iJO1366 model split into 103 isolated modules, with each module comprising nearby reversible reactions that can be assigned to a specific metabolic function or pathway (Fig 1, full list of modules is found in Table C in S1 Table). Seventy-three modules only contained a single structural reversible reaction, i.e., reversible reactions that are not constrained by surrounding irreversible reactions. In other words, around 70% of the modules are made by a single linear pathway of reversible reactions. Each of the six largest modules contained eight or more reversible reaction, representing the following metabolic pathways: (I) glycolysis, pentose phosphate pathway, ribonucleotides and sugars metabolism, (II) fatty acids metabolism, (III) pyruvate metabolism, (IV) nucleotides metabolism, (V) folate, serine and glycine metabolism, and (VI) TCA cycle (Table 1 and Fig 1).

In order to validate the biological relevance of the generated modules, we computed the correlation in gene expression between structurally reversible reactions of the *iJO1366* model belonging to the same module (intra module) and to different modules (extra module). Even without a modular structure, one would expect that proximity would enhance correlation. Hence, we calculated all correlations with distance two, i.e. those separated by exactly one reaction (Supporting text B in S1 Text and Table D in S1 Table). A distance of two was chosen because this is the shortest distance between two reversible reactions that belong to different modules. Correlation was computed using the PRECISE database [34], which contains 278 RNA-seq expression profiles for *E. coli* K-12 MG1655 and BW25113. Constitutively expressed genes (38 out of 164 genes presented in the GPR of reversible reactions), those with standard deviation of the normalized gene expression data less or equal to 0.6, as well as four genes that are part of the gene protein relationship (GPR) of 99 of the structural reversible reactions, were removed from the dataset. The latter four genes were removed in order to avoid artificially high correlations. Overall, 74% of the genes belonging to the GPR of the structural reversible reactions of iJO1366 were mapped onto the transcriptomic dataset.

If there is no modular organization underlying gene expression, we would expect no difference in the distance-2 correlation of structural reversible reactions between intra and extra module reactions. In contrast, we observed a significant difference in their distributions. The

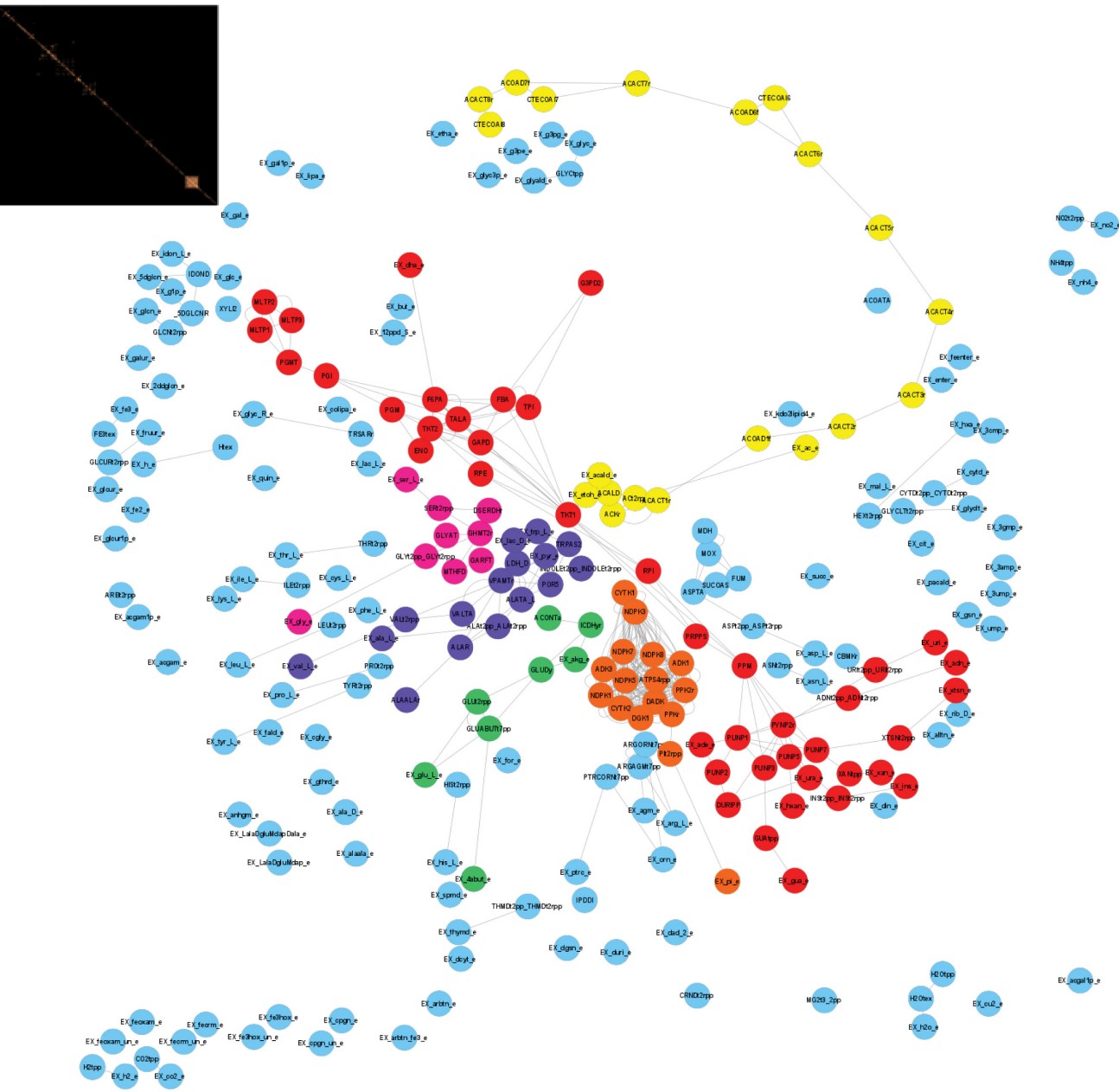

**Fig 1. Modules of structural reversible reactions of iJO1366.** The network was subdivided into modules of reversible reactions that share metabolites. Each node represents a reversible reaction and each edge a metabolite connecting two reversible reactions. The topology of the network is given by the invisible presence of irreversible reactions that connect the reversible reactions. The six largest modules that contain more than seven reversible reactions: TCA cycle (green), Fatty acids metabolism (yellow), Glycolysis, PPP, ribonucleotides and sugars metabolism (red), Pyruvate metabolism (purple), Folate, serine and glycine metabolism (pink), and Nucleotides metabolism (orange). Insert left corner shows the reaction adjacency matrix of the structural reversible reactions, which also represents the reversible reaction modules.

correlation between distance-2 reactions belonging to different modules was generally low with only 0.4% displaying absolute correlation values greater than 0.8 and 25% showing less than 0.1 absolute correlation value (Fig B in S1 Fig). In contrast, more than 37% of intra-module distance-2 reactions displayed an absolute correlation value greater than 0.8 and only 12%

**Table 1. Topologically independent modules of *iJO1366*.** The 294 structural reversible reactions were grouped in 103 modules (see Table C in S1 Table for the full list of modules). Here features of the six largest modules are presented.

| Module | Pathway | Number of Reactions | | | |
|---|---|---|---|---|---|
| | | Reversible | Irreversible Internal | Irreversible External | Total |
| 7 | TCA cycle | 8 | 5 | 26 | 39 |
| 11 | Fatty acids metabolism | 20 | 6 | 35 | 61 |
| 15 | Glycolysis, PPP, ribonucleotides and sugars metabolism | 42 | 25 | 147 | 214 |
| 18 | Pyruvate metabolism | 16 | 16 | 53 | 85 |
| 63 | Folate, serine and glycine metabolism | 9 | 11 | 32 | 52 |
| 86 | Nucleotides metabolism | 16 | 33 | 62 | 111 |

displayed an absolute correlation value less than 0.1. While there is substantial overlap in distributions, the two distributions are not only significantly different (p-value = $8.2 \times 10^{-59}$, Kolmogorov–Smirnov test for equal distributions), but the median of the intra module distribution is significantly higher than the extra module reactions (p-value = $1 \times 10^{-39}$, Wilcoxon sum rank test). We conclude that the module structure is biological relevant as reflected in differential gene expression.

## Human model Recon 2.2

Analogous to iJO1366 model, Recon 2.2 clustered into 251 isolated modules. However, 53% of the 1,582 structural reversible reactions in the reduced model were allocated to a single large cluster, describing a major share of the central carbon metabolism. Inspection of the topological features of this cluster revealed that the high connectivity was caused by a large number of antiporter reactions coupling metabolism of otherwise distinct metabolites. For example, Recon 2.2 contains 102 antiporter reactions catalysed by the L-type neutral amino acid transporter (LAT1) generated by pairwise combination of the possible substrates. These antiporters play a critical homeostatic role as "harmonizers", maintaining a balanced cytosolic pool of all amino acids [35]. In order to explore the intracellular metabolism, however, the majority of antiporter reactions (414 reactions) were removed from the model, keeping the uniporter transport reactions and the co-transport reactions that were needed to enable the transport of all metabolites in the model. Notably, this reduction of reversible antiporter reactions does not affect the overall model capabilities in terms of metabolites that the model is able to consume and produce and maximum specific growth rate (details of simulations in Supporting text A in S1 Text and Table E in S1 Table). Clustering of the reduced model produced 321 modules (list of modules in Table F in S1 Table), with the larger module containing 111 reversible reactions. Almost half of the modules (45%) contained only one structural reversible reaction (Fig A in

**Table 2. Topologically independent modules of Recon 2.2.** The remaining 1,168 structural reversible reactions, after removal of antiporter reactions, were grouped in 321 modules (see Table F in S1 Table for the full list of modules). Here features of the six largest modules are presented.

| Module | Pathway | Number of Reactions | | | |
|---|---|---|---|---|---|
| | | Reversible | Irreversible Internal | Irreversible External | Total |
| 4 | Fatty acids metabolism | 46 | 47 | 86 | 179 |
| 6 | Glutamate and glutathione metabolism and TCA cycle | 81 | 221 | 78 | 380 |
| 7 | Nucleotides metabolism | 111 | 119 | 63 | 293 |
| 9 | Pyruvate, lactate, alanine and cysteine metabolism | 38 | 99 | 59 | 196 |
| 19 | Glycolysis, PPP and galactose metabolism | 32 | 13 | 54 | 99 |
| 27 | Glycine, serine and taurine metabolism | 47 | 109 | 48 | 204 |

S1 Fig), while the six largest modules contained more than 30 reversible reactions each (Table 2 and Fig 3).

As with the iJO1336 model, the biological relevance of the Recon 2.2 model modules was validated by computing the Pearson correlation in gene expression data between the genes of structurally reversible reactions with distance 2 (Supporting text B in S1 Text and Table G in S1 Table). The human expression data from the GTEx (The Genotype-Tissue Expression) project [36] was used to compute gene correlations. This dataset contains RNA-seq data for 17,382 samples representing 54 different human tissues from 948 donors. Gene TPMs were obtained from the GTEx portal on 13/04/21 (dbGaP Accession phs000424.v8.p2). The data was initially filtered by removing genes with less than 10 samples with at least 1 TPM and normalized by log-transformation log2(TPM+1). Out of the 397 genes present in the reversible reactions of the Recon 2.2 model, only the gene expressing phosphoglycerate mutase 2 (HGNC:8889) was not found in the GTEx database.

Fig 2B shows the distribution of correlations between structurally reversible reactions of distance 2 from the Recon 2.2 model. The distribution of the correlation for extra module reactions resembled a normal distribution, centred at zero, with 53% of correlations presenting absolute values of less than 0.1 and only 10% displaying absolute correlation values higher than 0.5. In contrast, for the intra module correlations, the maximum absolute correlation frequency was between 0.55 and 0.6. Less than 30% of correlations were between -0.1 and 0.1 and more than 30% showed absolute correlation values over 0.5. Unlike the *E. coli* data, the correlation between human genes rarely exceeds 0.8, possibly due to the inherent expression variation between individuals compared to isogenic *E. coli*. While seemingly less profound, the difference between the distributions of correlation of intra and extra module genes is highly significant (p-value $< 5x10^{-324}$, Kolmogorov–Smirnov test for equal distributions) with a tendency of the genes belonging to the same module to have higher correlations (p-value $< 5x10^{-324}$, Wilcoxon sum rank test).

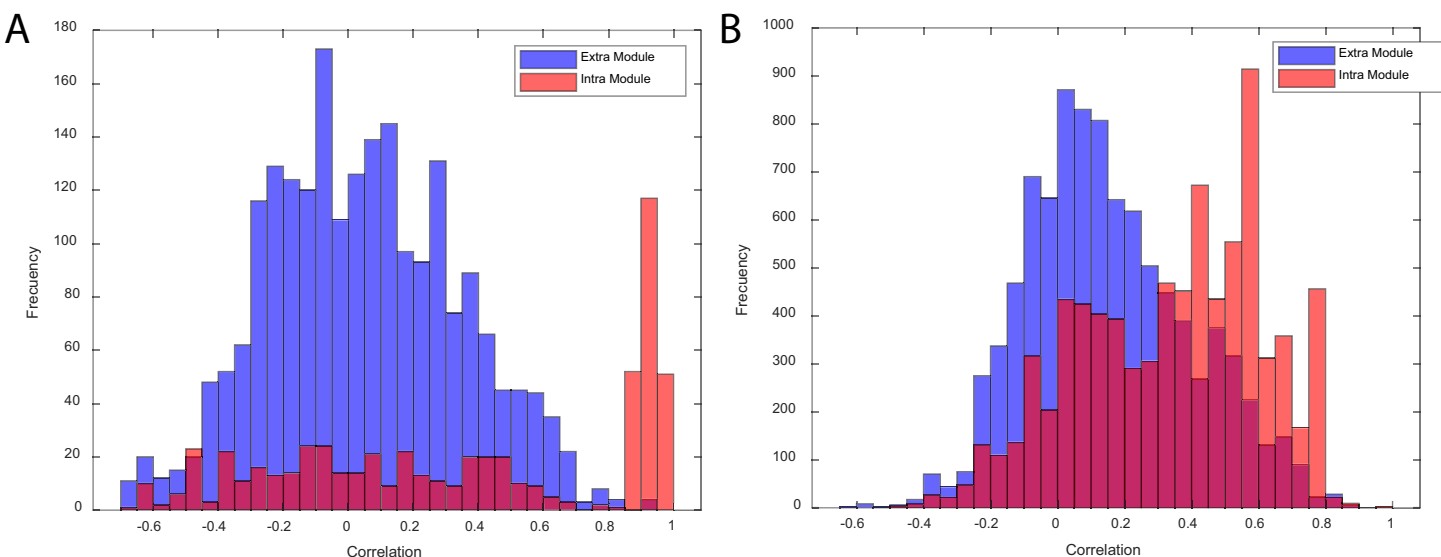

**Fig 2. Distribution of correlation between structural reversible reactions of distance 2. A.** Results for the *iJO1366* model. **B.** Results for the Recon 2.2 model. The distribution of the (absolute) correlations between the structural reversible reactions with distance 2 that belong to the same module (Intra Module) and to different modules (Extra Module), are shown in red and blue, respectively.

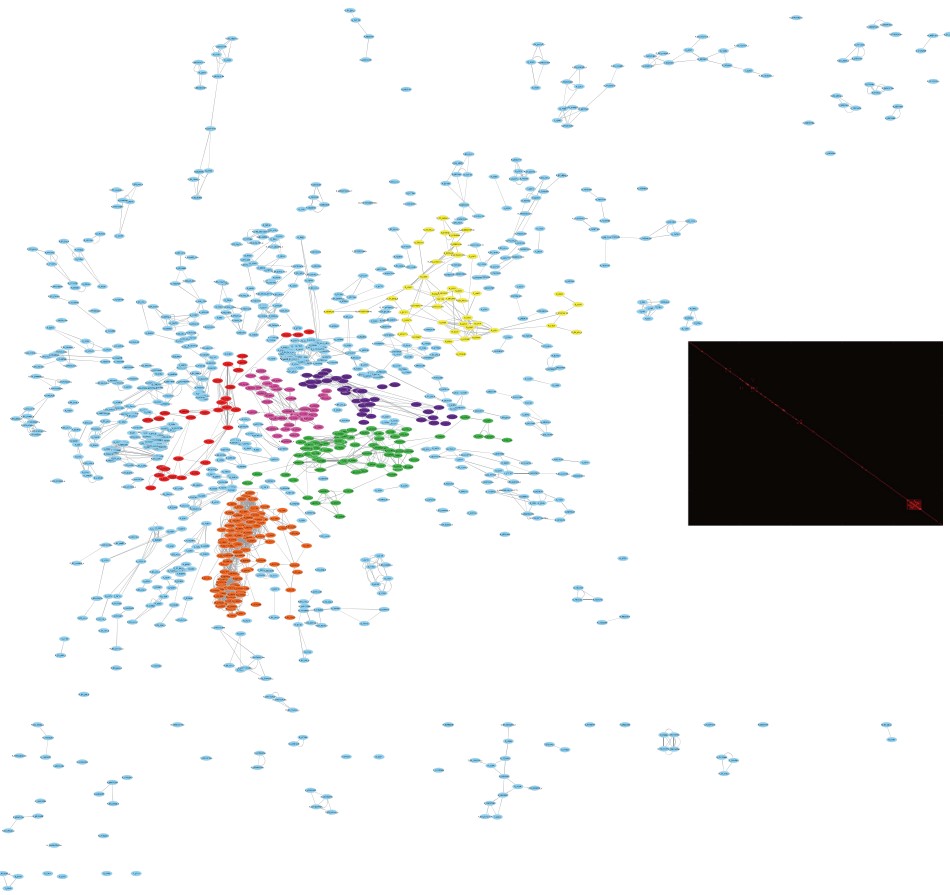

**Fig 3. Modules of structural reversible reactions of Recon 2.2 model.** The network was subdivided into modules of reversible reactions that share metabolites. Each node represents a reversible reaction and each edge a metabolite connecting two reversible reactions. The topology of the network is given by the invisible presence of irreversible reactions that connect the reversible reactions. The six largest modules that contain more than 30 reversible reactions: Glutamate metabolism, glutathione metabolism and TCA cycle (green), Fatty acids metabolism (yellow), Glycolysis, PPP, and galactose metabolism (red), Pyruvate, lactate, alanine and cysteine metabolism (purple), Glycine, serine and taurine metabolism (pink), and Nucleotides metabolism (orange). On the right the reaction adjacency matrix of the structural reversible reactions is presented, which also represents the reversible reaction modules.

## Topological flexibility in *iJO1366*

Irreversible reactions direct flux from substrates towards biomass components and products. Reversible reactions grant metabolic networks the flexibility to have alternative flow directions, e.g. glycolysis versus gluconeogenesis, in order to adapt to a changing environment [12]. We can characterize the level of flexibility by the possible directed topologies (DTs), defined as the set of distinct network configurations where all reversible reactions are unidirectional and all network reactions carry flux simultaneously at steady state.

The reduced and compressed *E. coli* model iJO1366 has 248 structural reversible reactions (not directionally constrained by surrounding irreversible reactions) (Table A in S1 Table), thus it can be described by a maximum of $2^{248}$ different DTs if these reactions were completely independent. In reality, reactions are coupled [31], and the number of distinct feasible DTs arising from the various combination of directions for each reversible reaction should be substantially less. We defined the "topological" degree of freedom (DoF) ($Log_2N$) of the metabolic network (N being the total number of feasible DTs), as the minimum number of reversible

reactions that must be directionally fixed in order to fix metabolism to a distinct feasible DT. From a practical point of view, it may be possible to identify the metabolic state of the cell under certain growth conditions by determining the direction of key reactions by the use of metabolomics data and thermodynamic principles [32, 33, 37–39], if the "topological" DoF is a relatively small number.

In order to quantify the flexibility of the iJO1366 model, the "topological" DoF was estimated. It is possible to estimate the feasibility of each DT by FVA, however, this task is computationally prohibitive for $2^{248}$ DTs. By taking advantage of the modular topology of metabolic models, where each module is semi-autonomous and consists of highly connected components, we can estimate an upper bound of the model "topological" DoF as the Cartesian product of the individual modules "topological" DoFs, greatly reducing the computation challenge.

The largest module contained n = 42 reversible reactions, therefore, $3.7 \times 10^{14}$ ($2 \, x \, n \, x \, 2^n$) optimization problems would have to be solved to determine the "topological" DoF of this module. As this is simply impractical, we constructed a set of rules to identify flux patterns leading to infeasible DTs, i.e., unable to carry flux at steady state (see Methods). The rules were constructed using patterns that either violate single metabolite mass balances (i.e., steady state) or the second law of thermodynamics by generating infeasible closed loops (Fig 4A and 4B). Finally, the "topological" DoF was estimated by performing a depth-first search in the directionality space of each module using the previously defined rules to remove infeasible DTs. Importantly, by using this topological search approach and in contrast to previous methods [23], no optimizations runs were required for the "topological" DoF estimation.

A comparison of this enumeration strategy against FVA for all but the largest module revealed the above two principles for rule generation captured almost all the infeasible DTs patterns. The majority of failures were found in metabolite pairs that behaved as one metabolite due to the reactions connecting them. This issue was found, for example, in the Arginine metabolism module, where Agmatine (agm[p]) and Arginine (arg_L[p]) are fully mixed by reactions connecting them (Fig 4C). This issue was overcome by introducing a local mass balance around the two fully mixed metabolites to generate the missing infeasible patterns (see Materials and Methods). A few failures remained in module 86 (nucleotides metabolism) even after the inclusion of the local mass balance rule; the search strategy identified 446 feasible DTs compared to 440 determined using brute force FVA. A heuristic approach was implemented to find the missing infeasible flux patterns needed to find the correct number of DTs. The heuristic approach generates potential new infeasible patterns based on the already identified infeasible patterns with 3 or more fixed directionalities, by moving the constraint from one reaction to another that was initially unconstrained (see Supporting text C in S1 Text for more details).

The inclusion of the heuristic approach, as well as identifying the six missing infeasible DTs on module 86, reduced the number of calculated feasible DTs on the largest module (module 15) by 43% (from $1.14 \times 10^{11}$ to $4.89 \times 10^{10}$), and the computational time by 88%. Due to the large number of structural reversible reactions in module 15 (46 structural reversible reactions after model compression, see Methods), it is infeasible to run the brute force search to fully validate our search algorithm. Instead, sampling was used to validate the enumeration approach for this module. Out of a random sample of $1 \times 10^7$ DTs analysed using FVA, only 0.055% (5,519) were feasible. This percentage is similar to the 0.069% DTs found feasible for module 15 using our search algorithm. Furthermore, all of the infeasible DTs seen in the random sample could be explained with the existing infeasible flux patterns. We complemented random sampling with a targeted sampling to ensure that for every combination of 5 structural reversible reactions, the full combinatorial within those reactions was covered. Out of the 43,864,128

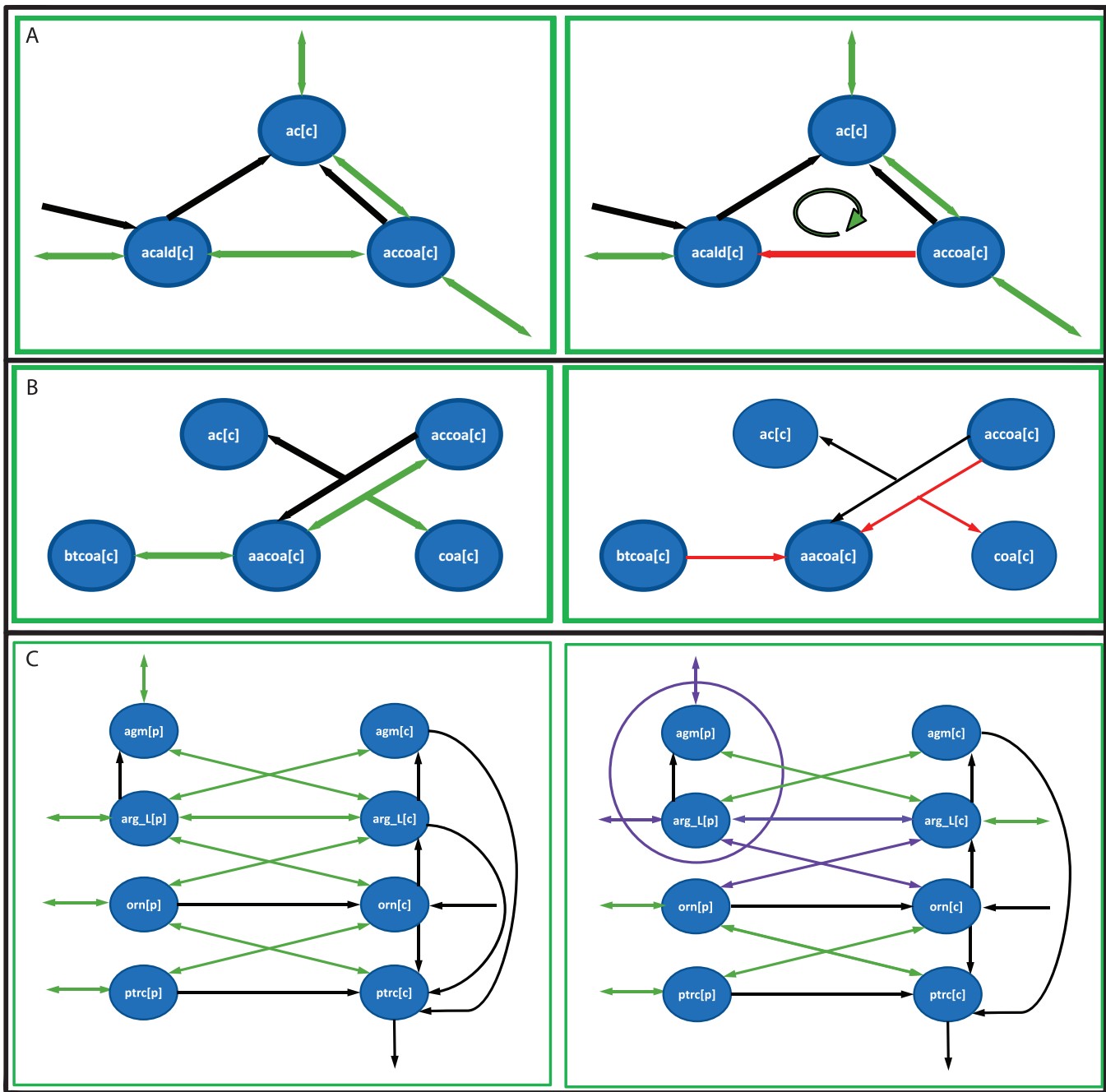

**Fig 4. Example of rules to identify infeasible flux patterns. A**. Loopless rules for the fatty acid metabolism module. In order to break the loop, the acetaldehyde oxidoreductase reaction (acetaldehyde (acald[c]) to acetyl-coA (accoa[c])) cannot point in the direction of acetaldehyde synthesis (red arrow). **B.** Local mass balance rules for the fatty acid metabolism module. Around each internal metabolite in the network, there must be at least one reaction in (synthesis) and one out (consumption). In this example, there is one irreversible reaction around acetoacetyl-coA in the synthesis direction, thus at least one of the reversible reactions should go in the consumption direction. **C.** Two metabolites that are fully mixed for the Arginine metabolism module. As Agmatine (agm[p]) and Arginine (arg_L[p]) are fully mixed, both can actually be seen as only one metabolite, thus a mass balance rule around both metabolites was added. Which means that around fully mixed metabolites there should be at least one reaction in (synthesis) and one out (consumption). In purple are highlighted the 3 reversible reactions around these metabolites. In red are presented the infeasible rules for A and B. Black arrows represent irreversible reactions and green arrows represent reversible reactions.

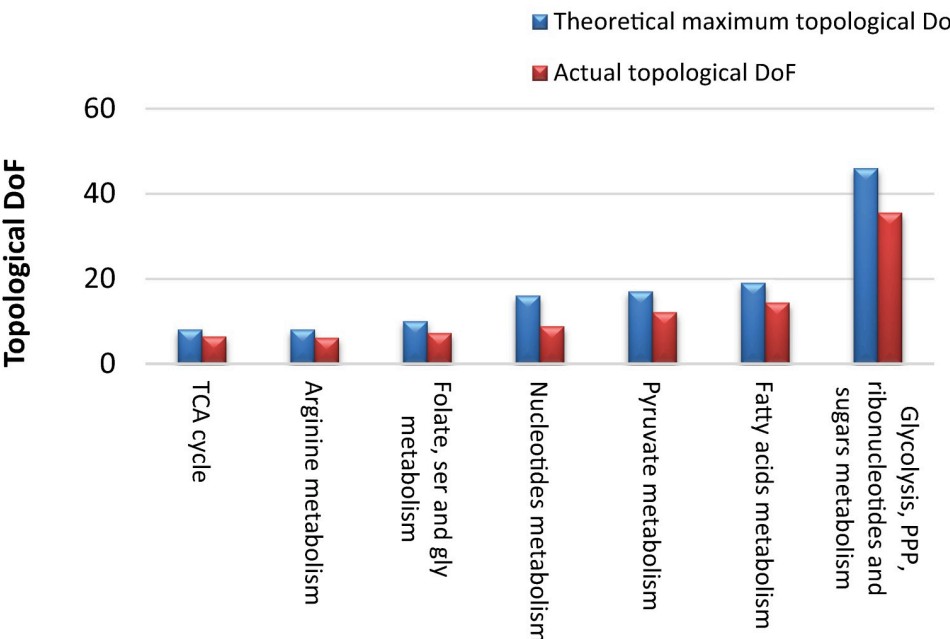

**Fig 5. Theoretical maximum and actual "topological" DoF of the seven larger modules of the *E.coli* iJO1366 reconstruction.**

($\frac{n!}{(n-5)!5!} x \, 2^5$, for n = 46 structural reversible reactions) simulations, 31,118 (0.071%) were found feasible. Again, the proportion of feasible DTs was very similar to the proportion found by our algorithm (0.069%), and the infeasible DTs were fully explained by the computed infeasible flux patterns. These results suggest that the presented topological enumeration algorithm is most likely capable of finding all the feasible DTs in unprecedentedly large search spaces.

For the largest module (module 15), there were 29 infeasible flux patterns identified: 11 due to loopless rules, 15 due to local mass balance rules, 2 due to two metabolites mixed mass balance rules and 2 from the heuristic algorithm (Fig B-H in S1 Fig). The infeasible patterns covered 38 of the 46 structural reversible reactions, leaving eight completely free reactions: G3PD, RPI, DURIPP, PUNP2, EX_ade_e, EX_dha_e, EX_hxan_e and EX_ura_e. Based on the number of feasible DTs, the "topological" DoF of module 15 is 35.5 out of the theoretical maximum 46 (Fig 5).

In order to gain a better understanding of the flexibility of structural reversible reactions within module 15, the directionality correlation between the reactions was studied (refer to Supporting text B in S1 Text for more details). Overall, the correlation between the studied reactions was poor, suggesting a weak coupling (Fig I in S1 Fig). We further investigated the presence of clusters of highly coupled reversible reactions within this module by enumerating the largest sets of fully connected reactions (maximal cliques) [40]. Here, connected reactions were defined as those with an absolute correlation higher than a predefined cut-off. When a cut-off of 0.85 was defined, only 5 cliques with 2 reactions each were found, each clique connecting an exchange and an internal reaction consuming the transported metabolite. The small number of cliques found and the extremely small size of them is in agreement with the previously found "topological" DoF of this module, confirming that the module structural reversible reactions are highly independent from each other.

The model "topological" DoF was estimated from the individual modules "topological" DoFs assuming that the modules are independent of each other. The assumption that reversible reactions within a module can operate independently of the state of reversible reactions in other modules largely appear to be valid because the structural reversible reactions in the model were organized in nearly isolated modules upon the removal of currency metabolites. Thus, changes inside a module are unlikely to have a high impact in a different module of the network [41]. The model "topological" DoF was estimated to be 200, i.e., 79% of structurally reversible reactions are independent of each other, which suggest a high level of flexibility. The simulation results of the seven largest modules on "topological" DoF, after model pre-processing, are presented in Fig 5.

## Discussion

Metabolic networks are inherently modular [19, 20, 22]. This modular nature provides a means for simplifying structural and functional analysis of large-scale metabolic networks. Early work described the network topology using an undirected graph with no consideration of the nature of the edges, hereby yielding artificial short path lengths and an ambiguous structure. By excluding currency metabolites and accounting for directionality of reactions, metabolic networks have been previously described having a bow-tie structure with a substrate subset connected to a product subset through a *giant strong component* corresponding to *central carbon metabolism* [20, 21].

The deliberate omission of energy/redox co-factors was critical for the identification of thermodynamically isolated modules. Clearly, these modules are coupled through energy and redox to other reactions, however, the coupling is to the tightly maintained global pool rather than between any two individual reactions. Arguably energy/redox homeostasis–maintaining energy charge and the ratios of various redox partners–is a more global regulatory principle (see for example [42]) than the modules identified. Conversely, assuming that coupling through energy/redox links individual reactions would speak against modularity, e.g., suggest that all gene regulation is globally coordinated with no modularity, which is clearly not accurate (operons in bacteria is a clear example of modularity). The thermodynamic isolation hereby employed focus on is the isolation achieved by irreversible reactions–commonly subject to allosteric regulation–that ensures the products have no impact on substrate concentrations or the reactions upstream of the substrates. Importantly, once modules have been indentified, currency metabolites (redox and co-factors) may be reincorporated to the respective reactions if desired. For example, a kinetic model of a module would typically include cofactors as fixed concentration external metabolites [43, 44]. It is for the sole purpose of identifying modules that currency metabolites are reversibly removed. Altogether, the identified modules unveiled the modular organization of the reversible reactions of the *E. coli* iJO1366 and the Recon 2.2 GeMs into metabolic modules connected by irreversible reactions. The resulting organization resembles conventional metabolic pathways and subsystems known for these organisms, but in this case, they emerged from the topological features of each network leveraged by a novel clustering approach. The approach unravelled hundred and three nearly isolated modules for the *E. coli* network growing aerobically in media with glucose as sole carbon source. The majority of the modules contained only one structural reversible reaction, whereas thirty contained more than one. For comparison, Ma and Zeng [20] found 29 strong components that include no less than three metabolites. When applying the clustering approach to a much larger reconstruction, Recon 2.2, a large number of antiporter transport reactions (e.g., the amino acid "harmonizers" such as LAT1) were removed, which artificially connect different parts of metabolism [35]. After the removal of antiporters, the model was subdivided into 321 thermodynamically isolated modules. The six largest modules in the

human model displayed known metabolic functions similar to those found in *E coli*, namely: TCA cycle, glycolysis, pentose phosphate pathway, fatty acids, nucleotides, sugars and amino acids metabolism.

The biological relevance of the identified modules was demonstrated through gene expression analysis, which showed that the correlation between of reversible reactions of distance 2 was significantly higher between reactions within a cluster and low between reactions in separate clusters. The majority of correlated enzymes catalysing reactions within the same module are highly correlated (more than 0.8 absolute value correlation across 278 transcription datasets in the *E.coli* model and more than 0.5 absolute value correlation across 17,382 RNA-seq samples in the Human model). In contrast, the majority of absolute correlations between reactions in different modules concentrated around zero supporting the presence of the inferred underlying modular structure.

The *E. coli* iJO1366 metabolic network flexibility was studied using the identified modules. An efficient depth-first search algorithm using simple infeasible mass balance and loopless rules was developed to explore the topological flexibility of the modules by enumerating all feasible DTs. We note that this amounts to enumerating all the flux topes in these (currency-free) subnetworks [23]. The analysis revealed only a weak coupling between structural reversible reactions in the largest module, which points to an overall high topological flexibility providing a high degree of robustness [45]. Strong coupling was only found between some boundary (exchange) and internal reactions consuming a common metabolite, which is known to be the case as exchange reactions can exert massive coupling and blocking of reactions at the boundary of metabolic networks [31].This observation is true across the modules. Assuming that modules operate independent of each other, the topological degree of freedom of the *E. coli* iJO1366 model was determined to be 200 (79%) out of a theoretical maximum of 248. This number represents an upper bound on the number of directionalities that must be determined to fix the topological state of the metabolic network. A more exact estimate would be obtained by enumerating all the feasible DTs in the entire network as whole, which is unfortunately impractical at this scale [23]. Still, we can conclude that except for linear pathways, reversible reactions operate practically independent of each other, granting both flexibility and robustness against internal and external perturbations [22, 34, 46, 47].

## Methods

### Model reduction and compression

Prior to model reduction and compression, the models were modified for mass balance and thermodynamic calculation consistency [32, 33]. In short, reactions catalyzed by different enzymes in either direction were lumped together, the species $HCO_3$, $CO_3$, $CO_2$, and $H_2CO_3$ were aggregated as $CO_{2tot}$ and $H_2O$ was added to the other side of the reaction. Additionally, the oxidized and reduced FAD of mitochondria were replaced by the oxidized and reduced FADenz to represent the enzyme-bound FAD cofactor. Finally, the models' original directionality constraints were used, which describe aerobic growth with glucose as main carbon source (Tables H and I in S1 Table).

Two steps of model reduction were performed:

1. Flux variability analysis (FVA), the minimum and maximum flux of all network reactions were estimated by linear programming and blocked reactions were identified. Blocked reactions are reactions that carry null flux as a result of being linked to singleton/dead-end metabolites, or caused by contradicting reversibility specifications. Additionally, reversible reactions that operate irreversible under the specified directionality constrains were

identified. The model was reduced by removing blocked reactions and constraining flux direction of the identified irreversible reactions.

2. Model compression, linear pathways were lumped into single reactions. Reactions organized into a linear pathway must carry the same flux value, therefore can be lumped together [48]. If one of the reactions in the pathway is irreversible the lumped reaction will be irreversible.

## Clustering approach

1. A novel approach to divide the metabolic network into functional modules (metabolic pathways) was developed. First, the currency metabolites (co-factors and moieties), defined as highly connected metabolites with a carrier function (e.g., electrons and chemical groups), were removed from the network reactions. The removal of currency metabolites was done manually based on on their role in each reaction. For instance, if a currency metabolite was participating as carrier in the reaction, it was removed, otherwise it was kept (refer to Table J in S1 Table for the complete list of removed metabolites). In addition, if it was not clear which metabolites were the co-factors and main substrates/products of the reaction (e.g., K+-Cl- cotransport: cl[e]+k[e] = cl[c]+k[c]), all metabolites were kept. In order to preserve the metabolic functions of the network, currency metabolites were kept in all reactions that synthetise or degrade them, as well as when a metabolite was being used as a building block (e.g. Acetoacetyl-CoA:acetate CoA-transferase: acac+coa = aacoa) (Tables K and L in S1 Table). Then reversible reactions were clustered by common metabolites connecting them. The irreversible reactions were kept in the model to retain the network structure, but these reactions did not take part of any module. The reaction adjacency matrix was used to reveal the generated modules.

## Model flexibility and enumeration of directed topologies

In order to study the degree of flexibility conferred to the metabolic model due to reversible reactions, two terms were introduced:

1. Directed Topologies (DTs), defined by fixing directions to all reversible reactions, i.e., all reactions in the network are irreversible and carry flux in a DT.

2. Topological degree of freedom, defined as $\log_2 N$ where N the number of DTs that are capable of carrying flux in all reactions.

The DT enumeration problem was divided into sub-problems each contained in a predefined module. Inside each module, duplicated reactions were lumped as one reaction and irreversible reactions in opposite direction were lumped into new reversible reactions. Then, the connections between reversible reactions inside the modules were studied using a set of rules that define infeasible flux patterns rendering the candidate DT infeasible. These rules are based on the mass balance around one or two metabolites and the loopless flux condition. Next, a depth-first search was performed through the feasible space of reversible reactions directions using the previously defined rules as a table of infeasible flux patterns (see next section for details on the rules and search algorithm). In order to reduce the computational time, the search algorithm was implemented in C, resulting in a 500 times faster search. To verify that the feasible space was properly explored by the search algorithm, an FVA brute force algorithm was implemented and their results were compared. It was found that for some of the largest modules, few infeasible patterns were missing, therefore, a heuristic method for finding

the missing infeasible flux patterns was added. For the largest module, a flux sampling strategy was employed for validation as the brute force method was impractical to run. For more information about the missing rules and the adjustment of the patterns check Supporting text A in S1 Text. Finally, simulations were executed in MATLAB (The Math-Works, Natick, USA) using the MEX file interface for C code. Gurobi Optimization solver was employed for the linear optimizations. Calculations were performed on a desktop computer.

## Depth-first search for DT enumeration

Enumeration of all DTs in a given module of a determined model is likely NP-hard. Even when directed topologies are considered and composed of configurations where all reactions carry flux (i.e. a *maximal* flux mode)–common simplification applied to keep the problem tractable [29]–, this enumeration amounts to counting all flux modes of size *K* (where *K* represents the size of the subnetwork), which can be assumed to be NP-complete from previous theoretical results [49]. Thus, implementation of a brute-force optimization algorithm where all the DTs are checked for feasibility is simply naïve and does simply not scale well.

In order to overcome this obstacle, we developed and implemented a novel depth-first search algorithm that traverses a *directionality tree*, where the current flux directionality pattern is compared against a list of previously calculated infeasible flux patterns for early rejection. A similar graph-based approach has been used to efficiently enumerate elementary flux modes using a *reaction tree* for rejecting *on-the-fly* pathways with two-futile cycles [50]. Particularly in our case, we constructed two types of infeasible flux patterns before each search in each module: 1) mass-imbalanced flux patterns, and 2) loopy flux patterns. The first infeasible pattern type is derived directly from inspection of the stoichiometric matrix, $\mathbf{S}$. Each row *j* of $\mathbf{S}$, denoted by $\mathbf{S}_j$, represents the mass balance for a metabolite *j* in the module, and more importantly, it provides necessary relations for reaching a mass balanced/imbalanced flux topology based on a simple criterion: *an imbalanced flux pattern has either all fluxes coming or leaving from metabolite j*. Such topologies will never reach steady state, and hence, they can be discarded without the need of simulation. Moreover, since $\mathbf{S}_j$ is typically sparse due to the removal of currency metabolites, exhaustive computation of all mass-imbalanced flux pattern combinations for all the metabolites in the module can be efficiently performed.

The second infeasible pattern type follows the same logic but addresses thermodynamically infeasible loops. The presence of internal loops or closed cycles in flux solutions violate the second law of thermodynamics, as they can potentially reach infinite flux without additional energy input [27]. In order to identify and correct such situations, the 'loopless´ flux optimization formulation has been proposed and successfully applied to large metabolic models [44, 51–53]. Briefly, thermodynamically infeasible flux solutions can be identified by checking if they contain any active (non-zero) closed cycles [54]. These closed cycles are encoded in the ´loop-law´ matrix, $\mathbf{N}_{int}$, which describes a null space basis of the stoichiometric matrix of internal reactions $\mathbf{S}_{int}$ of the network [53]. We have recently developed an efficient algorithm called Fast-SNP [44] for finding a sparse representation of such basis in large-scale metabolic models [55, 56], substantially easing the detection of closed cycles in flux patterns. Once $\mathbf{N}_{int}$ is computed for the particular submodule, then complete enumeration of all 'loopy' flux patterns in the module can be efficiently performed. It is important to highlight that $\mathbf{N}_{int}$ is constructed using the original reaction stoichiometries, thereby avoiding distortion of the generated loop laws.

## Supporting information

**S1 Text.  A: Checking model capabilities after removing antiporter reactions** (in S1 Text). **B: Calculation of correlation in gene expression between structurally reversible reactions**

of distance 2 (in S1 Text).**: Missing Infeasible Patterns** (in S1 Text). **D: Study of directionality correlation between structural reversible reactions** (in S1 Text).
(DOCX)

**S1 Fig. A: Characterization of Recon 2.2 modules dimensions.** Almost half of the generated modules contain one reversible reaction, while 6% of the modules contain 10 or more reversible reactions (in S1 Fig). **B: Infeasible flux patters of module 15 due to loopless rules.** 5 of the 11 infeasible flux patterns are presented, each with a different colour. The arrows represent the reactions in the module (both reversible and irreversible) and the nodes the metabolites (in S1 Fig). **C: Infeasible flux patters of module 15 due to loopless rules.** 4 of the 11 infeasible flux patterns are presented, each with a different colour. The arrows represent the reactions in the module (both reversible and irreversible) and the nodes the metabolites (in S1 Fig). **D: Infeasible flux patters of module 15 due to loopless rules.** 2 of the 11 infeasible flux patterns are presented, each with a different colour. The arrows represent the reactions in the module (both reversible and irreversible) and the nodes the metabolites (in S1 Fig). **E: Infeasible flux patters of module 15 due to mass balance rules.** 14 of the 15 infeasible flux patterns are presented, each with a different colour. The arrows represent the reactions in the module (both reversible and irreversible) and the nodes the metabolites (in S1 Fig). **F: Infeasible flux patters of module 15 due to mass balance rules.** 1 of the 15 infeasible flux patterns are presented, each with a different colour. The arrows represent the reactions in the module (both reversible and irreversible) and the nodes the metabolites (in S1 Fig). **G: Infeasible flux pattern of module 15 due to two metabolites mixed mass balance rules.** The infeasible pattern is highlighted using red arrows. The arrows represent the reactions in the module (both reversible and irreversible) and the nodes the metabolites (in S1 Fig). **H: Infeasible flux pattern of module 15 found using the heuristic algorithm.** The two infeasible patterns are highlighted using red and blue arrows. The arrows represent the reactions in the module (both reversible and irreversible) and the nodes the metabolites (in S1 Fig). **I: Correlation analysis of the directionalities of the structural reversible reactions in module 15.** (A) Absolute correlation distribution of the directionalities in the DTs sample. (B) Cliques distribution and (C) Cliques size distribution for different absolute correlation thresholds. The higher the absolute correlation cut off, the least and smaller the size of the cliques found (in S1 Fig).
(DOCX)

**S1 Table. A: Summary of reduction and compression of E.coli iJO1366 model.** First a singleton reduction was realized where blocked reactions and singleton metabolites were removed, and found new irreversibilities were added; then the model was compressed by lumping linear pathways (in S1 Table). **B: Summary of reduction and compression of Recon 2.2 model.** First a singleton reduction was realized where blocked reactions and singleton metabolites were removed, and found new irreversibilities were added; then the model was compressed by lumping linear pathways (in S1 Table). **C: Topologically independent modules of iJO1366.** The 294 structural reversible reactions were grouped in 103 modules (in S1 Table). **D: Correlation in gene expression between structurally reversible reactions of distance 2 of the iJO1366 model** (in S1 Table) **E: Effect of removing antiporter reactions.** Flux variability analysis shows that range fluxes do not differ between full [vminF,vmaxF] and reduced [vminR,vmaxR] model (in S1 Table). **F: Topologically independent modules of Recon 2.2.** The remaining 1,168 structural reversible reactions, after removal of antiporter reactions, were grouped in 321 modules (in S1 Table). **G: correlation in gene expression between structurally reversible reactions of distance 2 of the Recon 2.2 model** (in S1 Table) **H: Directionality constraints used for *E.coli* iJO1366 model** (in S1 Table) **I: Directionality constraints used for *Recon 2.2* model** (in S1 Table) **J: Currency metabolites that were**

**removed from reactions** (in S1 Table). **Table K: Reversible reactions with and without currency metabolites of _E.coli_ iJO1366 model** (in S1 Table). **Table L: Reversible reactions with and without currency metabolites of Recon 2.2 model** (in S1 Table) **Table M: Cliques distribution for different correlation thresholds in the largest metabolic modules of the iJO1866 model** (in S1 Table).
(XLSX)

**S1 Data. Code to calculate correlations and visualise the data.**
(ZIP)

**S2 Data. Correlations computed for Human Recon 2.2.** Code to process large TPM files from the GTEx (The Genotype-Tissue Expression) project and produce correlation tables.
(ZIP)

## Author Contributions

**Conceptualization:** Verónica S. Martínez, Pedro A. Saa, Lake-Ee Quek, Lars K. Nielsen.

**Data curation:** Verónica S. Martínez, Pedro A. Saa, Kanupriya Tiwari.

**Formal analysis:** Verónica S. Martínez, Pedro A. Saa.

**Funding acquisition:** Lars K. Nielsen.

**Investigation:** Verónica S. Martínez, Pedro A. Saa, Lars K. Nielsen.

**Project administration:** Lars K. Nielsen.

**Software:** Jason Jooste.

**Supervision:** Lars K. Nielsen.

**Writing – original draft:** Verónica S. Martínez.

**Writing – review & editing:** Verónica S. Martínez, Pedro A. Saa, Lake-Ee Quek, Lars K. Nielsen.

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
