## [Decision Letter · Decision Letter 0]

24 Sep 2021

Dear Dr. Nielsen,

Thank you very much for submitting your manuscript "The topology of genome scale reconstructions unravel independent metabolic modules and high model flexibility" (PCOMPBIOL-D-21-01428) for consideration at PLOS Computational Biology. As with all papers peer reviewed by the journal, your manuscript was reviewed by members of the editorial board and by several independent peer reviewers. Based on the reports, we regret to inform you that we will not be pursuing this manuscript for publication at PLOS Computational Biology.

In line with the points raised by the reviewers (please see below), the omission of energy/redox co-factors is a fundamental shortcoming towards identifying modules that are thermodynamically isolated.

The reviews are attached below this email, and we hope you will find them helpful if you decide to revise the manuscript for submission elsewhere. We are sorry that we cannot be more positive on this occasion. We very much appreciate your wish to present your work in one of PLOS's Open Access publications. 

Thank you for your support, and we hope that you will consider PLOS Computational Biology for other submissions in the future.

Sincerely,

Tunahan Cakir

Guest Editor

PLOS Computational Biology

Kiran Patil

Deputy Editor

PLOS Computational Biology

Reviewer's Responses to Questions

**Comments to the Authors: **

Reviewer #1: We have read “The topology of genome scale reconstructions unravel independent metabolic modules and high model flexibility” by Martinez et al. The authors explored the topology of metabolic network reconstructions by defining sets of thermodynamically isolated modules connected by irreversible reactions. The modules corresponded with correlations in RNA transcription data. A major focus was on Directed Topologies which were defined by fixing the direction of reversible reactions and then determining how many could carry flux. The authors explored these topologies for metabolic networks of E. coli and Human metabolism. Overall, the manuscript describes a nice approach for defining metabolic subsystems in an unbiased way. However, we have several questions regarding the methods and assumptions used by the authors. Unfortunately, the methods section lacking leading to several questions related to the robustness of the method.

Major Comments:

• Our major concern relates to some ambiguity in the methods section. The authors describe two steps of model reduction. The first uses FVA to estimate the flux of all network reactions. However, the authors do not specify the constraints applied to their models prior to FVA. For the E. coli model, were the FVA calculations run using default constraints (e.g. aerobic M9 minimal media + glucose?) or was this a fully open model? What about for the human reconstruction? Changing constraints, especially exchange reactions will change FVA results. Thus, we must know what constraints were used. 

• Do the authors anticipate dramatically different results from altering these constrains? Would the identified modules change dramatically? Is a sensitivity analysis needed?

• The authors casually discuss and dismiss several concerns without describing the basis for such statements or being more specific in what they mean. For example, on line 234 the authors state that “reduction of reversible anitiporter reactions don’t affect overall model capabilities”. How are the authors defining model capabilities? How was this determined? We understand that they likely mean capabilities in terms of catabolic and anabolic capabilities, but what about gene essentiality or other model predictive capabilities? The authors should be careful with their language and statements like these, and may need to demonstrate that no other model capabilities are changed based on removing such reactions.

• We commend the authors use of RNA transcription data to bolster their results. However, its difficult for us to validate these results without deposition of data (both RNA transcription data and the model simulation results). We understand that the transcription data is from a separate published study but the outcome of the filtering steps (e.g. describe on line 199) are important to evaluate.

Minor Comments:

• Line 59: add “of metabolism” – reconstructions can be constructed for several other systems

• Line 61 – “not”

• Line 270 “visually” may not be the correct word

• We encourage another round of grammar checks to address issues such as those above

Reviewer #2: The work by Martinez et al aims to identify well defined metabolic modules in genome-scale metabolic models. To this end the authors focus on metabolically connected reversible reactions that are separated by irreversible reactions form each other. Each of these reversible cluster defines one model. 

The question of identifying modules in metabolism is very interesting and — in my understanding — has not been satisfactorily been answered so far. The approach by the authors is interesting. I do have, however, some concerns that relate to the strategy taken by the authors and the stability of the identified modules.

* The authors remove currency metabolites from the analysis. This is a key step as it is removes many links. Thus it needs to be clearly defined what abundance makes a metabolite a currency metabolite. If the threshold that separates currency metabolites from non-currency metabolites is changed, do the cluster remain the same?

* Removing links (currency metabolites and anti porters) is the key for the success of the presented method. In fact previous attempts to identify modules in (genome-scale) metabolic networks that did not remove links only succeed in identifying modules for the optimal flux cone, but not for the general, see https://doi.org/10.1007/s00285-013-0731-1 and https://doi.org/10.1038/srep00580 Thus I believe that removing links needs to be more thoroughly justified. In particular the removal of anti-porters appears to me as an adhoc assumption to make the method work. Also I encourage the authors to discuss similarities and differences between the modules identified by the authors to the optimal modules identified in the mentioned articles. 

* A nice feature of the work by Kelk et al (see above) is that the modules identified by Kelk et al. can nicely explain the combinatorial explosion of elementary flux modes (EFMs) in large metabolic networks, as they originate from the combination of the EFMs in the modules. Can such an argument be translated to the modules identified here as well?

* With respect to the feasible reversible flux space identified by the authors, what is its relation to the reversible flux cone constraint by the set of minimal metabolic behaviors (MMB, see https://doi.org/10.1016/j.dam.2008.06.039). I think, they are the same. If so, what is the novelty in the proposed concept?

* I do not understand if the authors assume a particular medium or if they take the models with all exchange reactions unconstrained. In the later case I again wonder if similar modules arise as for a specific medium I would expect that many reversible reactions turn irreversible resulting in a much smaller topological flexibility within a fixed environment. This also touches on the validation with the expression data of 54 different human tissues. Why do the authors not use tissue specific genome-scale reconstructions for this analysis?

* The authors argue that their method identifies thermodynamically isolated modules and that reversible reactions are highly independent of each other. I like to disagree that these modules are isolated and that reversible reactions are highly independent of each as thermodynamic coupling is still possible via currency metabolites. For instance, the reversible Glutamate dehydrogenase in E. Coli turns into a thermodynamic bottleneck if grown on glucose, see https://doi.org/10.1128/jb.176.15.4664-4668.1994

* The question of how many directed topologies exist within a given environment has been theoretically addressed in terms of so called flux topes, see https://doi.org/10.1093/bioinformatics/bty550 I think it would be interesting to compare the authors heuristics with the exhaustive method—at least on a theoretical level. 

* The authors develop a heuristics to compute directed topologies an suggest that with their algorithm it is likely to find all feasible directed topologies. I think this statement needs to be explicitly shown. Either in a smaller network, like coli_core, http://www.asmscience.org/content/journal/ecosalplus/10.1128/ecosalplus.10.2.1#backarticlefulltext, or ECC2, https://doi.org/10.1038/srep39647, or by restricting the analysis to the optimal flux cone in a large network.

Reviewer #3: The work by Martínez et al focuses on the well-known problem (re)interpretation of metabolic networks, finding modules. This work focuses on thermodynamic perspective, reversibility of reactions and topological properties of some (sub) networks. The novelty of the proposed method is questionable, if at all unclear. 

The idea seems interesting at first yet there are several points to be addressed in this paper, in particular the novelty is questionable and the criteria is rather poorly described and is unclear what is the difference between earlier and proposed concepts. The following points need to be addressed before any decision.

P16, L279. The authors quantify the flexibility by (number of?) possible directed topologies. The authors need to elaborate this criterium, as this already has been tackled by the elementary modes concept. How this metric is different than earlier defined concepts?

When the currency metabolites are removed from the system, what happens to the reactions that consumes/produces these? The reason to focus on this is as follows: the presence of internal loops or closed cycles can be drawn from the analysis of the stoichiometric matrix (e.g. analysis of null space as already mentioned in the manuscript) which is somewhat “distorted” by the removal of these currency metabolites. 

Focusing on finding the properties of the sub-network, how the proposed method is different from elementary flux modes (or extreme pathways) problem is not clear, and judging from enumeration approach accompanied by the classical analysis of stoichiometric matrix, it seems that this is study a minor contribution to the field.

**Have the authors made all data and (if applicable) computational code underlying the findings in their manuscript fully available?**

Reviewer #1: No: Supplemental data appears to be missing and should be included in a revision

Reviewer #2: No: * I may have overlooked it, but I did not see a link to the actual scripts and models used in the main text.

Reviewer #3: None

PLOS authors have the option to publish the peer review history of their article (what does this mean?). If published, this will include your full peer review and any attached files.

Reviewer #1: No

Reviewer #2: No

Reviewer #3: No

---

## [Decision Letter · Decision Letter 1]

8 Apr 2022

Dear Dr. Nielsen,

Thank you very much for submitting your manuscript "The topology of genome-scale metabolic reconstructions unravels independent modules and high network flexibility" for consideration at PLOS Computational Biology. As with all papers reviewed by the journal, your manuscript was reviewed by members of the editorial board and by several independent reviewers. The reviewers appreciated the attention to an important topic. Based on the reviews, we are likely to accept this manuscript for publication, providing that you modify the manuscript according to the review recommendations.

In line with the points raised by the two reviewers, please (i) include the criteria you used in classifying a metabolite as currency metabolite to remove it from the network, and (ii) explain the logic behind using log2 in determining topological degrees of freedom. Please submit a revised version that address those points.

Sincerely,

Tunahan Cakir

Guest Editor

PLOS Computational Biology

Kiran Patil

Deputy Editor

PLOS Computational Biology

[LINK]

In line with the points raised by the two reviewers, please (i) include the criteria you used in classifying a metabolite as currency metabolite to remove it from the network, and (ii) explain the logic behind using log2 in determining topological degrees of freedom. Please submit a revised version that address those points.

Reviewer's Responses to Questions

**Comments to the Authors:**

Reviewer #1: The authors have satisfactorily addressed all of our comments

Reviewer #2: Thank you, authors, for your detailed response and the modifications to your manuscript.

My concerns have been sufficiently addressed.

I still wonder about the definition of cofactors and the impact of this definition on your results. Especially, I regret that you decided not to test the sensitivity of your results on a slightly different definition of cofactors (e.g. by increasing our decreasing the number of required links that make a metabolite a cofactor).

At the least I'd ask you to make your current statement more precise "defined as highly connected metabolites were removed from the network reactions. The removal of currency metabolites was done manually based on the nature of the metabolites in each reaction. In order to preserve the metabolic functions of the network, currency metabolites were kept in reactions that synthesize or degrade consume them (Supporting Tables S10 and S11)." Please could you include the minimum number of links that makes a metabolite removable, (and any additional guiding principles that where used in your manual curation) For instance, in case there are 2 alternative synthesis pathways, did you keep both in or just one? etc.

As for your ability on "isolated thermodynamic modules" I remain skeptical. Particularly I disagree with your response that "There is no thermodynamic coupling between the two assimilation systems" with respect to GS/GOGAT. Previously, we found that there is a coupling transmitted via cofactors, see https://doi.org/10.1038/srep08930 I'd love to discuss these aspects with you in future. However, this disagreement does not hinder publication.

Reviewer #3: The revised work by Martínez et al focuses on determination of directed topologies, finding modules from thermodynamics perspective. There are several major changes in the article and the new version is clearly improved version.

My earlier comments were centered on three points:

- metrics in enumerating different modules.

- difference with earlier work

- fate of currency metabolites

These points (fate of currency metabolites to a much lesser extent) have been addressed by the authors.

Finally, the use of log2 in determining the degree of freedom needs to be motivated.

**Have the authors made all data and (if applicable) computational code underlying the findings in their manuscript fully available?**

Reviewer #1: Yes

Reviewer #2: Yes

Reviewer #3: Yes

PLOS authors have the option to publish the peer review history of their article (what does this mean?). If published, this will include your full peer review and any attached files.

Reviewer #1: No

Reviewer #2: **Yes: **Jürgen Zanghellini

Reviewer #3: No

Figure Files:

Data Requirements:

Reproducibility:

References:

---

## [Editor Report · Decision Letter 2]

14 May 2022

Dear Dr. Nielsen,

We are pleased to inform you that your manuscript 'The topology of genome-scale metabolic reconstructions unravels independent modules and high network flexibility' has been provisionally accepted for publication in PLOS Computational Biology.

Best regards,

Tunahan Cakir

Guest Editor

PLOS Computational Biology

Kiran Patil

Deputy Editor

PLOS Computational Biology

---

## [Editor Report · Acceptance letter]

9 Jun 2022

PCOMPBIOL-D-21-01428R2 

The topology of genome-scale metabolic reconstructions unravels independent modules and high network flexibility

Dear Dr Nielsen,

I am pleased to inform you that your manuscript has been formally accepted for publication in PLOS Computational Biology. Your manuscript is now with our production department and you will be notified of the publication date in due course.

With kind regards,

Zsofia Freund
